# Neutrophil-to-Lymphocyte Ratio and Early Variation of NLR to Predict In-Hospital Mortality and Severity in ED Patients with SARS-CoV-2 Infection

**DOI:** 10.3390/jcm10122563

**Published:** 2021-06-09

**Authors:** Laure Abensur Vuillaume, Pierrick Le Borgne, Karine Alamé, François Lefebvre, Lise Bérard, Nicolas Delmas, Lauriane Cipolat, Stéphane Gennai, Pascal Bilbault, Charles-Eric Lavoignet

**Affiliations:** 1Emergency Department, Regional Hospital of Metz-Thionville, 57000 Metz, France; l.abensurvuillaume@chr-metz-thionville.fr (L.A.V.); l.cipolat@chr-metz-thionville.fr (L.C.); 2Emergency Department, Hôpitaux Universitaires de Strasbourg, 67000 Strasbourg, France; karine.alame@chru-strasbourg.fr (K.A.); pascal.bilbault@chru-strasbourg.fr (P.B.); 3INSERM (French National Institute of Health and Medical Research), UMR 1260, Regenerative NanoMedicine (RNM), Fédération de Médecine Translationnelle (FMTS), University of Strasbourg, 67000 Strasbourg, France; 4Department of Public Health, University Hospital of Strasbourg, 67000 Strasbourg, France; francois.lefebvre@chru-strasbourg.fr; 5Emergency Department, Haguenau Hospital, 67500 Haguenau, France; lise.berard@ch-haguenau.fr; 6Emergency Department, Colmar Hospital, 68000 Colmar, France; nicolas.delmas@ch-colmar.fr; 7Emergency Department, Reims University Hospital, 51000 Reims, France; sgennai@chu-reims.fr; 8Emergency Department, Hôpital Nord Franche Comté, 90400 Trévenans, France; charles-eric.lavoignet@hnfc.fr

**Keywords:** COVID-19, NLR, severity, mortality

## Abstract

(1) Introduction: The neutrophil-to lymphocyte ratio is valued as a predictive marker in several inflammatory diseases. For example, an increasing NLR is a risk factor of mortality in sepsis. It also appears to be helpful in other settings such as cancer. The aim of our work was to study the prognostic value of NLR for disease severity and mortality in patients infected with SARS-CoV-2 upon their admission to the Emergency Department (ED) and its early variation (ΔNLR) in the first 24 h of management (H-24). (2) Methods: Between 1 March and 30 April 2020, we conducted a multicenter and retrospective cohort study of patients with moderate or severe coronavirus disease 19 (COVID-19), who were all hospitalized after presenting to the ED. (3) Results: A total of 1035 patients were included in our study. Factors associated with infection severity were C-reactive protein level (OR: 1.007, CI 95%: [1.005–1.010], *p* < 0.001), NLR at H-24 (OR: 1.117, CI 95%: [1.060–1.176], *p* < 0.001), and ΔNLR (OR: 1.877, CI 95%: [1.160–3.036], *p*: 0.01). The best threshold of ΔNLR to predict the severity of infection was 0.222 (sensitivity 56.1%, specificity 68.3%). In multivariate analysis, the only biochemical factor significantly associated with mortality was again ΔNLR (OR: 2.142, CI 95%: ([1.132–4.056], *p*: 0.019). The best threshold of ΔNLR to predict mortality was 0.411 (sensitivity 53.3%; specificity 67.3%). (4) Conclusion: The NLR and its early variation (ΔNLR) could help physicians predict both severity and mortality associated with SARS-CoV-2 infection, hence contributing to optimized patient management (accurate triage and treatment).

## 1. Introduction

For over a year now, the world has been witnessing a global health crisis caused by an emerging coronavirus called severe acute respiratory syndrome coronavirus 2 (SARS-CoV-2) which was first identified in Wuhan, China [1]. The Greater-East region of France was one of the pandemic’s epicenters in Europe, where, as of mid-February 2021, this emerging virus resulted in nearly 7800 deaths and infected over 47,582 patients.

Although the pathophysiology of COVID-19 is not yet fully elucidated, it has been demonstrated that it correlates with a pro-inflammatory state. Studies have shown that this infection is associated with a dysregulation of the immune response, particularly inducing T lymphopenia with a frequent decrease in CD4+ T cells. This decrease in lymphocytes is more pronounced in severe SARS-CoV-2 infection due to the aggravated inflammatory response and production of a cytokine storm [2,3]. Thus, lymphopenia has often been regarded as the reference biomarker to judge the severity of COVID-19 [4]. This infectious disease, like other inflammatory states, is also followed by an increase in neutrophils [2]. However, neutrophilia, along with lymphopenia, is not specific for severity or mortality assessment in SARS-CoV-2 infection [5,6]. The NLR appears to be a good reflection of the systemic inflammatory response to a pathogen and may be more useful than lymphocytes or neutrophils alone in assessing the prognosis of patients with COVID-19 [7,8,9,10].

An increasing NLR is a risk factor of mortality in sepsis, where it can be a useful prognostic biomarker in critically ill patients [11,12]. This predictive marker appears to be helpful in other settings such as cancer [13]. Moreover, in emergency care, NLR appears to be a greater predictor of bacteriemia compared to other routine parameters such as C-reactive protein (CRP) level and lymphocyte and neutrophil counts [14]. Recently, several studies have reported that patients with severe COVID-19 disease had a higher NLR value compared to patients with moderate disease [15,16]. In a recent meta-analysis of 13 studies involving 1579 patients, Li et al. [17] reported that NLR had sensitivity and specificity values of, respectively, 78% and 85% for disease severity and sensitivity and specificity values of 83% each for disease mortality. Hence, NLR could be fully integrated as a severity and mortality predictor in the management of infected patients both in the Emergency Department (ED) and in the subsequent hospital stay. However, the use of NLR variation, especially in the early stages of patient management, does not seem to have been subjected to study before.

Our aim was to investigate the prognostic value of NLR for disease severity and mortality in patients infected with SARS-CoV-2, upon their admission to the ED and in the first 24 h of management (H-24), excluding those who might have other confounding factors affecting their blood counts.

## 2. Methods

### 2.1. Study Population and Settings

We conducted a retrospective multicentric study in six ED of the Northeast region of France. We led our study in two university hospitals (CHRU of Strasbourg and CHU of Reims) and four general hospitals (Colmar Hospital, Nord Franche-Comté Hospital, Metz-Thionville Hospital and Haguenau Hospital). These hospital centers, along with the entire Greater-East region of France, constituted one of the outbreak’s epicenters in Europe during the first wave.

We included all adult patients who were hospitalized for COVID-19 after presenting to the ED between 1 March and 30 April 2020. All patients included in our study had a laboratory-confirmed diagnosis of COVID-19 by RT-PCR on nasopharyngeal swab. We excluded patients who had a non-confirmed diagnosis, those who received outpatient care, and those who received palliative therapy or limitation of therapeutic effort upon admission to the ED. Patients with a medical history or treatment that altered their blood counts and, therefore, their circulating lymphocytes or neutrophils (e.g., chemotherapy, immunosuppressive therapy, long- and short-term corticosteroid therapy, pre-admission antibiotic therapy, active cancer, or hematological malignancies) were also excluded from our study.

### 2.2. Data Collection

We retrospectively queried patients’ electronic medical records for epidemiological, clinical, and biochemical data and then standardized the results in a report file. We recorded symptom onset date along with patient’s current treatment and medical history (including cardiovascular disease, diabetes, pre-existing renal failure, cancer, and hematological diseases). The primary endpoint was the prognostic value of NLR on in-hospital mortality. The secondary endpoint was its prognostic value on severity of disease, where severe disease was defined by patient admission to the ICU (patients under invasive mechanical ventilation), and moderate disease was defined by patient admission to conventional hospitalization units (most patients with oxygen therapy). Ambulatory patients were excluded. Obesity was defined by a body mass index superior to 30 kg/m^2^. Functional autonomy was measured by the Knaus score. Standard biochemical parameters, such as levels of creatinine, CRP, total leukocytes and lymphocytes, were also collected. Lastly, we measured lymphocyte and neutrophil count and NLR early variation (ΔNLR), i.e., the difference between NLR values at H-24 and upon admission to the ED.

### 2.3. Ethics

This study was approved by the local ethics committee of the University of Strasbourg in France (reference CE: 2020-39), which, in accordance with the French legislation, waived the need for informed consent of patients whose data were entirely retrospectively studied.

### 2.4. Statistical Analysis

The statistical analyses included a descriptive section and an analytical section. We performed a descriptive analysis of the categorical variables by providing the frequency of each value along with the cumulative frequency. We performed a descriptive analysis of the continuous variables by providing location parameters (mean, median, minimum, maximum, first and third quartiles) and dispersion parameters (standard deviation, variance, range, and interquartile range). Normality of the distributions was tested using a normality test, such as the Shapiro–Wilk or Kolmogorov–Smirnov test, and was assessed graphically using a normal quantile plot. Comparisons between categorical variables were performed using Chi-squared test or Fisher’s exact test in case of expected values below 5 in any of the cells of the contingency table. Comparisons between continuous and categorical variables were assessed using the Student’s t-test or Wilcoxon’s test in case of heteroscedasticity or if the variable did not follow a normal distribution. Multivariate analyses were performed using all relevant variables. Multivariate analysis was performed using all variables obtained at admission or at H-24. Variables were selected using a backward stepwise method based on the Akaike Information Criterion to retain the relevant variables. The significance level was set at 5%. All the statistical analyses were generated with R 4.0.2. The cut-off of ΔNLR was defined using the Youden’s index. We created a validation cohort within our cohort by dividing it into two parts to confirm and justify our data (Appendix A). Both cohorts were created using a random draw method.

## 3. Results

### 3.1. Characteristics of the Study Population

During the study period, a total of 49,326 patients were admitted to the ED of all six hospitals. Of these patients, 4470 had a laboratory-confirmed SARS-CoV-2 infection and, in fine, 1035 patients were included in our study (Figure 1).

Our cohort had a median age of 69 ((58–79)) years and was predominately male (58.8%, CI 95%: (55.8–61.8)). One-third of the study population was obese (34%). In terms of medical history, over half of the patients (56.7%) had high blood pressure, over a quarter of them (26.7%) had a history of diabetes, and 23.2% of them presented pre-existing renal failure. At admission, the median NLR was significantly higher in the group presenting severe disease compared to that with moderate disease (5.2 ((3.2–8.7) versus 6.6 (4.1–11.1), *p* < 0.001). Our findings were similar at H-24 (4.4 (2.7–7) versus 7.4 (4.7–12.5), *p* < 0.001). Thus, over half (58%, CI 95%: (52–64.3)) of the severely affected patients presented a positive ΔNLR. Principals clinical and biochemical patient characteristics are summarized in Table 1.

### 3.2. Biochemichal Factors Associated with Severe COVID-19

Of the total study population, 789 patients (76.2%) had moderate disease, whereas 246 (23.8%) had severe disease, which required ICU management. In multivariate analysis, the factors associated with the severity of the infection were CRP (OR: 1.007, CI 95%: (1.005–1.010), *p* < 0.001), NLR at H-24 (OR: 1.117, CI 95%: (1.060–1.176, *p* < 0.001), and ΔNLR (OR: 1.877, CI 95%: (1.160–3.036); *p*: 0.01). These values are summarized in Table 2.

### 3.3. Predictive Factors of Severe COVID-19

We determined two ROC curves to predict the risk of disease severity. Regarding NLR at admission, the area under the curve (AUC) was 0.593 (CI 95%: (0.552–0.634), *p* < 0.001). The best cutoff for predicting the risk of infection severity was 6.883; it yielded a sensitivity of 48.3% (CI 95% (41.9–54.8)) and a specificity of 65.6% (CI 95% (62.2–69.0)). In multivariate analysis, if NLR was greater than 6.883, the OR was valued at 0.975 (CI 95%: (0.947–1.005), *p*: 0.097).

Regarding ΔNLR, the AUC was 0.627 (CI 95%: (0.580–0.674), *p* < 0.001). The best cutoff for predicting the risk of infection severity was 0.222; it provided a sensitivity of 56.1% (CI 95% (49.2–62.9)) and a specificity of 68.3% (CI 95% (64.4–72.1)). In multivariate analysis, if ΔNLR was greater than 0.022, the OR was valued at 2.187 (CI 95%: (1.348–3.547), *p*: 0.002). These results are consistent with those of the validation cohort (Appendix A) and are presented in Figure 2.

### 3.4. Biochemical Factors Associated with Mortality

A total of 139 patients died during their hospital stay, representing 13.4% (CI 95%: (11.4–15.5)) of our cohort. Upon admission to the ED and at H-24, the NLR values were associated with mortality in univariate analysis. However, the only biochemical parameter significantly associated with mortality in multivariate analysis was ΔNLR (OR: 2.142, CI 95%: (1.132–4.056), *p*: 0.019). These results are summarized in Table 3.

### 3.5. Predictive Factors of Mortality

Regarding NLR at admission, the AUC was 0.621 (CI 95%: (0.571–0.672)). The best NLR threshold for predicting the risk of death was 8.23; it yielded a sensitivity of 47.4% (CI 95%: (38.8–56.2)) and a specificity of 71.9% (CI 95%: (68.8–74.9)). If NLR was greater than 8.23, multivariate analysis was not found significant, with an OR valued at 1.025 (CI 95%: (0.989–1.063), *p*: 0.176). Regarding ΔNLR, the AUC was 0.576 (CI 95% (0.509–0.643)). The best ΔNLR threshold for predicting the risk of death was 0.411; it yielded a sensitivity of 53.3% (CI 95% (43.3–63.1)) and a specificity of 67.3% (CI 95% (63.7–70.8)). In multivariate analysis, if ΔNLR was greater than 0.411, then the OR was valued at 2.71 (CI 95%: (1.404–5.245); *p*: 0.003). These results are presented in Figure 3 and are consistent with those of the validation cohort (Appendix A).

### 3.6. Validation Cohort

The AUC of the training and validation cohorts were respectively 0.598 and 0.591 to predict critical illness and 0.629 and 0.616 to predict intra-hospital mortality, which validates the models. All data regarding the validation and training cohorts are available (Appendix A).

## 4. Discussion

The main endpoint of our study was to investigate the prognostic value of NLR in a cohort of patients infected with SARS-CoV-2, upon their admission to the ED and in the first 24 h of management (H-24). We selected patients as rigorously as possible to minimize all confounding factors that could alter the WBC. Our study confirmed that NLR is a potential marker to discriminate severity and mortality in patients with SARS-CoV-2 infection. In addition, we demonstrated the relevance of using ΔNLR to predict severity and mortality regarding this emerging disease.

Our results reflect other studies on pulmonary infections, where the NLR level upon admission to the ED was predictive of severity and mortality more accurately than other conventional biomarkers such as CRP [5,6]. De Jager et al. [5] established, in a cohort of 395 patients, an AUC of 0.701 for NLR, compared to 0.565 for CRP, 0.681 for neutrophils alone, and 0.672 for lymphocytes alone. In a prospective clinical study on 195 elderly subjects with acute pulmonary infection, Cataudella et al. [18] reported that NLR was an excellent predictor of mortality at 30 days, along with a correlation between the level of NLR and mortality (50% mortality for 13.4–28.3 NLR and 100% mortality if higher than 28.3 NLR). Concerning SARS-CoV-2 infection, NLR was also independently associated with progression to critical illness [19,20,21,22]. This association was demonstrated by Lian and al. [19] who also reported that the relationship between NLR and disease progression was significant and graded (HR:1.16 (CI 95%: (1.10–1.22); *p* < 0.001)). Moreover, NLR appears to be a predictive marker of mortality in this infection, as found in the meta-analysis of Li and al. [17], which included 10 studies and 2967 patients with sensitivity, specificity, and AUC values of 0.83, 0.83, and 0.90 respectively. These studies, although highlighting the prognostic value of NLR, did not take into account certain patient-related parameters potentially affecting the lymphocyte formula and, de facto, the NLR, nor did they precisely define the time of NLR measurement and its variation. Conclusively, our results stand more explicit: we refined our cohort by excluding patients who might have other causes modifying their lymphocyte formula and we clearly defined NLR measurements in detail. Therefore, our findings bring a novel and potential better view over the effect of SARS-CoV-2 on white blood cells (WBC) count.

Accordingly, there is great interest in considering NLR for its prognostic value in SARS-CoV-2 infection. Regarding clinical severity, we were able to demonstrate a significant difference in NLR upon admission to the ED when comparing moderate and severe infections. The difference between the two was even more relevant at H-24, which brings forward a marker that is specifically and significantly associated with severity and mortality, i.e., ΔNLR, which is the difference between NLR values at H-24 and upon admission to the ED. In our cohort, positive ΔNLR, signifying an increase in NLR between admission and H-24, was associated with a poor prognosis. Ye et al. [9] found similar results for a cohort of 349 patients, with an increase in NLR during hospitalization significantly associated with mortality. Riché et al. observed a reversed correlation in patients with septic shock [23]. Studies have indicated various thresholds to NLR [17] and, similarly to our findings, an NLR median value of 4.5 is often described in severe SARS-CoV-2 infections, with a recently demonstrated specificity of 86% and sensitivity of 74% [24]. On the other hand, in a population of 12,862 COVID-19 patients, Cai et al. [25] described an NLR threshold of 6.11, with sensitivity and specificity values of, respectively, 76% and 87% for predicting mortality, hence implying that, above this threshold, the introduction of corticosteroid therapy significantly reduced mortality.

One of the major keys in the management of this outbreak is controlling the number of patients in the ICU, which is the main reason behind the oversaturation of healthcare systems, an overload that increases mortality [26]. Therefore, the orientation of a COVID-19 patient between a conventional hospitalization unit or the ICU is essential. The value of the NLR could be coupled with that of other markers in a multi-marker approach or in a combined severity prediction score; this might be the key allowing optimization and standardization of patient triage in the ED.

### Limitations

Our study presents several limitations. First, it is retrospective, which means that, although we added a number of exclusion criteria (notably, comorbidities modifying the blood count and, therefore, the circulating lymphocyte and neutrophil count), the data are subject to other confounding factors. These data were collected during first wave of the pandemic, and at that time, the vast majority of our patients did not receive corticosteroid treatments modifying their WBC count, which are nowadays one of the cornerstones of severe SARS-CoV-2 infection management. Similarly, many patients were admitted to the ED after receiving non-recommended antibiotics that also affect the WBC count; these patients were also excluded from the study. Second, the ΔNLR may have been influenced by other factors related to patient management that we did not consider in the analysis, for lack of information that could not be exhaustively detailed. Decisively, the retrospective nature of the study did not allow us to extensively control the existence of other confounding factors such as bacterial co-infections, which could have allowed us to further refine our results.

## 5. Conclusions

The NLR and its early variation (ΔNLR) could help physicians predict both severity and mortality associated with SARS-CoV-2 infection, hence contributing to optimized patient management (accurate triage and treatment).

## Figures and Tables

**Figure 1 jcm-10-02563-f001:**
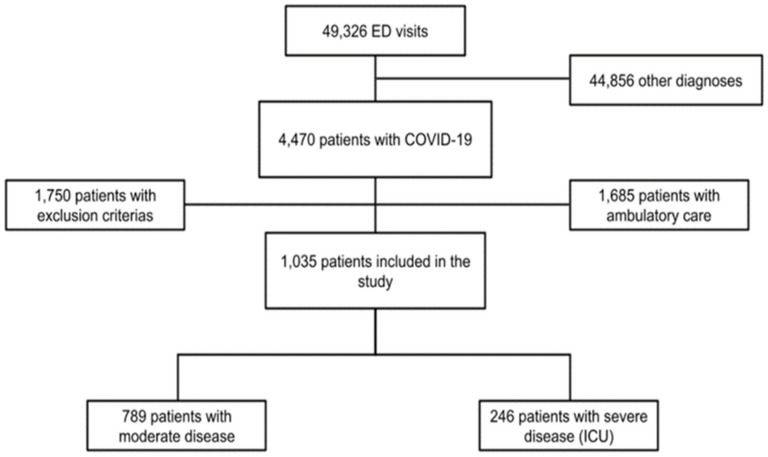
Flowchart of the study. Abbreviations: ED: Emergency Department, ICU: intensive care unit.

**Figure 2 jcm-10-02563-f002:**
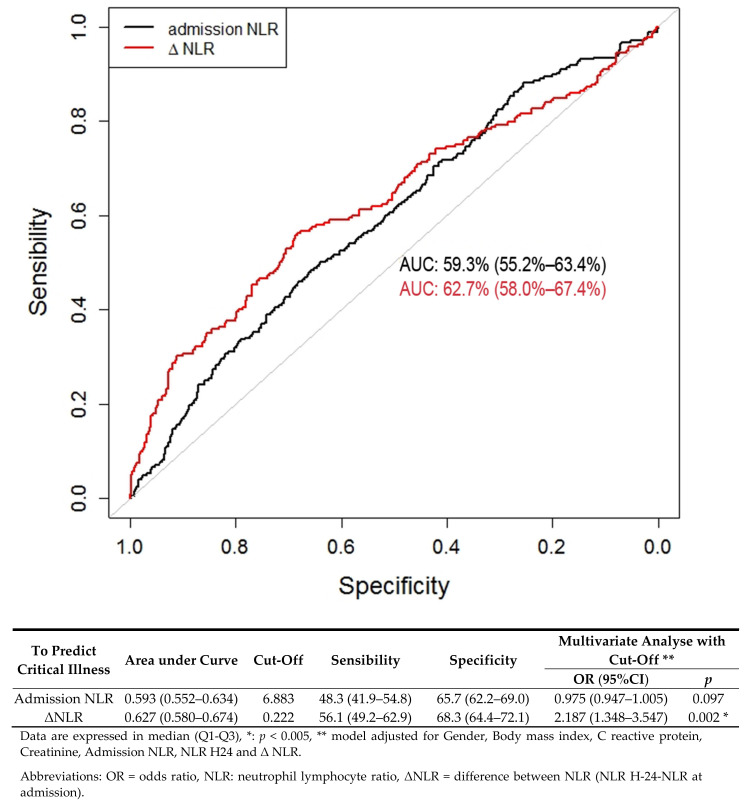
Receiver operating characteristics (ROC) curve for NLR as a predictive factor of severe COVID-19.

**Figure 3 jcm-10-02563-f003:**
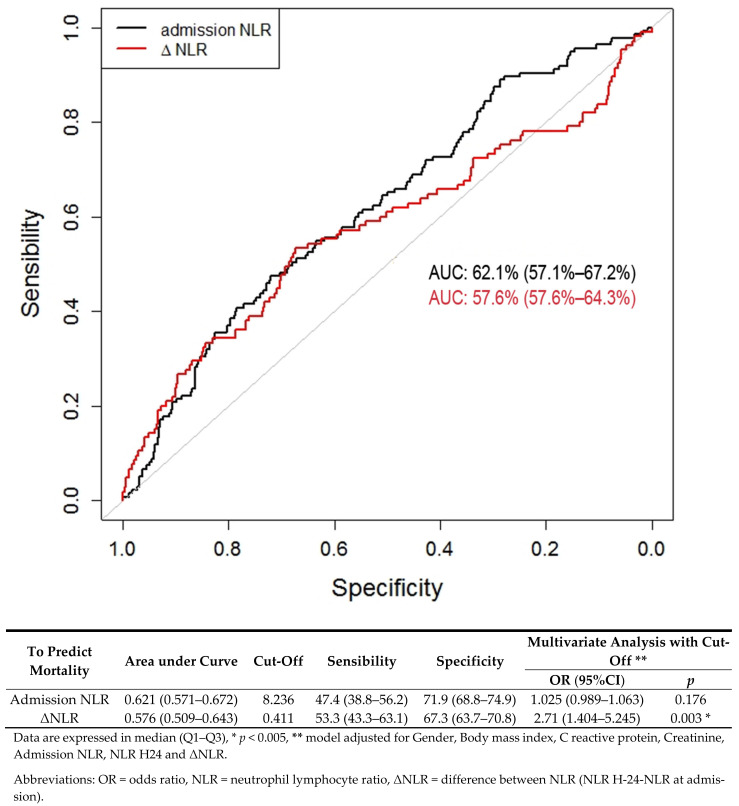
Receiver operating characteristics (ROC) curve for NLR as a predictive factor of mortality in COVID-19 patients.

**Table 1 jcm-10-02563-t001:** Demographic, baseline, and laboratory characteristics of patients with COVID-19.

	All Patients *n* = 1035	Moderate *n* = 789	Severe *n* = 246	*p*
**General Characteristics**				
Age (years)	69 (58–79)	70 (58–81)	66 (57.3–72)	<0.001 *
Gender male	609 (58.8)	433 (54.9)	176 (71.5)	<0.001 *
Obesity (BMI ≥ 30)	281 (36.9)	193 (35.0)	88 (41.9)	0.076
**Chronic Medical Illness**				
Hypertension	587 (56.7)	453 (57.4)	134 (54.5)	0.416
Diabetes mellitus	275 (26.7)	202 (25.6)	73 (26.6)	0.207
CKD	237 (23.2)	199 (25.5)	38 (15.8)	0.002 *
Cardiovascular illness	357 (34.5)	291 (36.9)	66 (26.8)	0.004 *
Total Autonomy	796 (77.2)	569 (72.4)	227 (92.7)	<0.001 *
Respiratory illness	203 (19.6)	151 (19.1)	52 (21.1)	0.490
**Laboratory Findings**				
CRP (mg/L)	81 (39–142.3)	68 (33–128)	124 (76–192)	<0.001 *
Lymphocyte (×10^9^/L)	870 (630–1200)	900 (640–1220)	780 (590–1122)	0.003 *
Lymphocyte H24 (×10^9^/L)	940 (670–1300)	1010 (710–1360)	800 (570–1110)	<0.001 *
Neutrophil (×10^9^/L)	4930 (3430–6932)	4730 (3370–6620)	5510 (3760–8160)	<0.001 *
Neutrophil H24 (×10^9^/L)	4680 (3300–6765)	4395 (3005–6175)	6010 (4130–8210)	<0.001 *
Admission NLR	5.4 (3.5–9.3)	5.2 (3.2–8.7)	6.6 (4.1–11.1)	<0.001 *
H-24 NLR	5 (3.1–8.2)	4.4 (2.7–7)	7.4 (4.7–12.5)	<0.001 *
ΔNLR >0 (%)	334 (41.3)	211 (35.3)	123 (58.0)	<0.001 *
Δ NLR	−0.64 (−2.88–1.29)	−0.88 (−3.09–0.79)	0.48 (−1.78–3.72)	<0.001 *
**Outcome**				
Hospital stay (days)	10 (7–17.3)	8 (6–12)	24 (17–38)	<0.001 *
Intra-hospital mortality	139 (13.6)	82 (10.4)	57 (24.1)	<0.001 *

Data are expressed in median (Q1–Q3) *n* (%), where *n* is the total number of patients with available data. * *p* < 0.05. Abbreviations: BMI = body mass index, CKD = Chronic kidney disease, ED = Emergency Department, O2 = oxygen, °C = Celsius degree, CRP = C reactive protein, NLR= neutrophil to lymphocyte ratio, ΔNLR = difference between NLR (NLR H-24-NLR at admission).

**Table 2 jcm-10-02563-t002:** Biochemical factors associated with severe COVID-19 (admission to the ICU).

				Multivariate Analysis **	
	All	Moderate	Severe	OR (95% CI)	*p*-Value
CRP	81 (39–142.3)	68 (33–128)	124 (76–92)	1.007 (1.005–1.010)	<0.001 *
Admission lymphocytes	870 (630–1200)	900 (640–1220)	780 (590–1122.5)	1000 (1.000–1.000)	0.841
Admission NLR	5.4 (3.5–9.3)	5.2 (3.2–8.7)	6.6 (4.1–11.1)	0.971 (0.940–1.004)	0.082
H-24 NLR	5 (3.1–8.2)	4.4 (2.7–7.0)	7.4 (4.7–12.5)	1.117 (1.060–1.176)	<0.001 *
ΔNLR >0 (%)	334 (41.3)	211 (35.3)	123 (58.0)	1.877 (1.160–3.036)	0.010 *

Data are expressed in median (Q1–Q3) or *n* (%), where *n* is the total number of patients with available data. * *p* < 0.05, ** model adjusted for gender, Body mass index, C reactive protein, Creatinine, Admission NLR, NLR H24 and ΔNLR. Abbreviations: OR = odds ratio, BMI = body mass index, CRP = C reactive protein, NLR = neutrophil to lymphocyte ratio, ΔNLR = difference between NLR (NLR H-24-NLR at admission).

**Table 3 jcm-10-02563-t003:** Biochemical and predictive factors of mortality in COVID-19 patients.

			Univariate Analysis		Multivariate Analysis **	
	Survivors	Non-Survivors	OR (95% CI)	*p*-Value	OR (95% CI)	*p*-Value
CRP	100.0 (56–158)	78.5 (37–139)	1.003 (1.001–1.005)	0.006 *	1.002 (0.999–1.005)	0.276
Admission lymphocytes	720 (500–1000)	890 (650–1120)	0.999 (0.999–1)	0.004	1.000 (0.999–1.001)	0.967
Admission NLR	7.6 (4.3–11.8)	5.2 (3.3–8.8)	1.023 (1.006–1.042)	0.010 *	1.020 (0.987–1.055)	0.232
H-24 NLR	7.9 (5.1–14.1)	4.6 (2.9–7.5)	1.059 (1.035–1.084)	<0.001 *	1.006 (0.963–1.050)	0.798
Δ NLR >0 (%)	271 (39.0)	58 (55.2)	1.931 (1.277–2.920)	0.002 *	2.142 (1.132–4.056)	0.019 *

Data are expressed in median (Q1–Q3) or *n* (%), where *n* is the total number of patients with available data. * *p* < 0.05, ** model adjusted for Gender, Body mass index, C reactive protein, Creatinine, Admission NLR, NLR H24 and Δ NLR. Abbreviations: OR = odds ratio, BMI = body mass index, CRP = C reactive protein, NLR = neutrophil to lymphocyte ratio, ΔNLR = difference between NLR (NLR H-24-NLR at admission).

## Data Availability

All data analyzed as part of the study are included.

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
