# Peer review of "Neutrophil-to-Lymphocyte Ratio and Early Variation of NLR to Predict In-Hospital Mortality and Severity in ED Patients with SARS-CoV-2 Infection"

_jcm, 2021, doi:10.3390/jcm10122563_

Round 1

Reviewer 1 Report

The authors retrospectively assessed NLR and delta NLR as predictors of severe disease in COVID-19. The strength of the study is a relatively high number of patients included and relatively high level of clinical details provided, but the reviewer has following concerns:

  1. The specificity and sensitivity detected by the authors is relatively low, compared to other reports that addressed the NLR and COVID-19 severity. For instance, sensitivity of 0.76 with specificity of 0.87 was detected in a report by Jingjing Cai et al, Cell Methabolism, 2021: https://doi.org/10.1016/j.cmet.2021.01.002. In addition, the number of patients included in this study was more than 12.000, which is 10 times more than in the present study. The reviewer suggests to have a look at this paper and include it as a relevant reference, and discuss it.

  1. Validation cohort is missing. One way to overcome this would be by including a training data set (a subset of the present cohort), followed by analyses of the rest of the patients.

  1. Definition and inclusion/exclusion to the different severity groups should be clarified.

Minor comments:

  1. How was the cut-off chosen? Was Youden index considered?

  1. It might be helpful to present some of the data provided in the tables as figures, e.g. parameters presented in Table 2.

Author Response

Reviewer #1

The authors retrospectively assessed NLR and delta NLR as predictors of severe disease in COVID-19. The strength of the study is a relatively high number of patients included and relatively high level of clinical details provided, but the reviewer has following concerns:

The specificity and sensitivity detected by the authors is relatively low, compared to other reports that addressed the NLR and COVID-19 severity. For instance, sensitivity of 0.76 with specificity of 0.87 was detected in a report by Jingjing Cai et al, Cell Methabolism, 2021: https://doi.org/10.1016/j.cmet.2021.01.002. In addition, the number of patients included in this study was more than 12.000, which is 10 times more than in the present study. The reviewer suggests to have a look at this paper and include it as a relevant reference, and discuss it.

We are thankful and appreciative of Reviewer #1’s comments and suggestions.
As explained in the Methods section of our article "Patients with a medical history or treatment that altered their blood counts and therefore circulating lymphocyte or neutrophils (e.g. chemotherapy, immunosuppressive therapy, long and short-term corticosteroid therapy, pre-admission antibiotic therapy, active cancer or hematological malignancies) were excluded from our study.” This choice of exclusion was decided to limit as much as possible the confounding bias with respect to parameters that have an effect on the WBC, and therefore, the NLR. We were very strict with our inclusion and exclusion criteria as evidenced by the number of excluded patients. Other cohorts described in the literature did not make this selection and this could explain, at least in part, the differences observed in terms of sensitivity and specificity.

We have developed on this in the Introduction section, saying:

« The use of NLR variation, especially in the early stage of patient management, does not seem to have been studied before. (...) Excluding those who may have other factors affecting their blood counts. »

Furthermore, we meticulously studied the report suggested by Reviewer #1 and included it in our Discussion:

" On the other hand, in a population of 12,862 COVID-19 patients, Cai et al. (25) described an NLR threshold of 6.11, with a sensitivity and specificity values of, respectively, 76% and 87% on predicting mortality. Hence implying that above this threshold, the introduction of corticosteroid therapy significantly reduced mortality.” 

Validation cohort is missing. One way to overcome this would be by including a training data set (a subset of the present cohort), followed by analyses of the rest of the patients.

As advised by both Reviewers #1 and #2, we performed a validation cohort. Our cohort was split into training and validation sets. Given the superposition of the curves, our results were thus established.

We added the following explanatory details:

  • In the Methods section: “We created a validation cohort within our cohort by splitting it into two parts to validate our data (Supplemental data).”
  • In the Results section: “These results were consistent with those of the validation cohort (Supplemental data).”

Figure S1

Figure S2

Definition and inclusion/exclusion to the different severity groups should be clarified.

According to the WHO classification of severity of SARS-CoV-2 infection, severe disease is defined by patient admission to the ICU, and mild or moderate disease is defined patient admission to conventional hospitalization units. Ambulatory patients were excluded. We have clarified this point in the Methods section.

Minor comments:

- How was the cut-off chosen? Was Youden index considered?

The cut-off has been chosen with the Youden’s index. We added this detail in the Methods section.

- It might be helpful to present some of the data provided in the tables as figures, e.g. parameters presented in Table 2.

We understand this position and we have seriously considered providing these figures. Nevertheless, it seems to us that this second table would be complex to extrapolate as a figure.

Reviewer 2 Report

I would like to thank the authors for the effort made to advance our knowledge of this infection. However, I consider that the work has some shortcomings that should be addressed:

Major concerns

Introduction

- As acknowledged in the introduction, there are meta-analysis data on the use of NLR. What is new in this work?

Methods:

- The endpoint of the study is not defined in the method. An unspecific endpoint is stated at the end of the introduction and in the title, but is not specifically defined in the methods. How is mild, moderate, and severe disease defined? Was mortality assessed? Over what time frame?

- The predictive model used has many shortcomings. The method of variable selection is not specified: was it performed after previous univariate exploratory analyses? This would not be correct from a methodological point of view. Is there any data on the validation of the model?

- According to Table 1 there are many variables with significant differences between groups, why does the multivariate model not include them?

Results:

- In the first paragraph describing the sample I do not understand the use of 95% CI.

- Table 1 can be simplified. The laboratory data do not provide too much relevant information (value creating a simplified table and putting the full table as Supplementary Material) How do you define moderate and severe disease? Why have these categories been chosen to present the data of the sample? It only makes sense to divide the table into these groups if the endpoint would include a comparison between these groups. 

- The presentation of results is cumbersome and difficult to interpret. I recommend summarizing the presentation of results to generate a clear and direct message.

Discussion:

- The AUR found with the model is low. The usefulness of the model should be discussed and discussed against previous literature in greater depth.

- Limitations should be expanded on, mainly focusing on the bias due to the retrospective data collection (it has been named), the inclusion of only hospitalized patients, and any data on other confounding variables, such as bacterial coinfections?

Minor concerns

- The introduction should be improved to focus on the problem to be solved and what this work contributes in this field.

- The English and style should be revised.

Author Response

Reviewer #2

I would like to thank the authors for the effort made to advance our knowledge of this infection. However, I consider that the work has some shortcomings that should be addressed:

Major concerns

Introduction

As acknowledged in the introduction, there are meta-analysis data on the use of NLR. What is new in this work?

We are grateful for Reviewer #2’s inquiries and comments

To answer the first concern, our study presents several new points:

  • The selection criteria of the patients in our cohort, allowing us to refine the possible confounding factors; we were very strict with our inclusion and exclusion criteria as evidenced by the number of excluded patients.
  • The accuracy of the time of measurement of NLR (at admission, H-24 and discharge)
  • The original contribution of the Δ-NLR

In order to make that point clear, we added the following details to the Introduction section:

“ (…) The use of NLR variation, especially in the early stage of patient management, does not seem to have been studied before.

In our study, we aimed to investigate the prognostic value of NLR on disease severity and mortality, in patients infected with SARS-CoV-2, excluding patients who may have other factors affecting their blood counts, upon their admission to the ED and at the first twenty-four hours of management (H-24).”

Then we added this precision to the Discussion section:
“The main purpose of our study was to investigate the prognostic value of NLR in a cohort of patients infected with SARS-CoV-2, upon their admission to the ED and at the first twenty-four hours of management (H-24). We selected patients as rigorously as possible to minimize all confounding factors that could alter the WBC.”

Methods:

- The endpoint of the study is not defined in the method. An unspecific endpoint is stated at the end of the introduction and in the title, but is not specifically defined in the methods. How is mild, moderate, and severe disease defined? Was mortality assessed? Over what time frame?

We thank you for these pertinent remarks. We have added clarifications in the Methods section in order to define these different elements:

“The primary endpoint was the prognostic value of NLR on in-hospital mortality. The secondary endpoint was its prognostic value on severity of disease, where severe disease was defined by patient admission to the ICU, and moderate or mild disease was defined patient admission to conventional hospitalization units.”

Regarding mortality (as mentioned in the outcome section of Table 1), we focused on in-hospital mortality and were unable to obtain any additional data after patient discharge.

- The predictive model used has many shortcomings. The method of variable selection is not specified: was it performed after previous univariate exploratory analyses? This would not be correct from a methodological point of view. Is there any data on the validation of the model?
As advised by bother Reviewers #1 and #2, we performed a validation cohort. Our cohort was split into training and validation sets. Given the superposition of the curves, our results were thus established.

We added the following explanatory details:

  • In the Methods section: “We created a validation cohort within our cohort by splitting it into two parts to validate our data (Supplemental data).”
  • In the Results section: “These results were consistent with those of the validation cohort (Supplemental data).”

Figure S1

Figure S2

- According to Table 1 there are many variables with significant differences between groups, why does the multivariate model not include them?
As described in the statistical section, we used a step-by-step selection method, which explains why there are many variables with significant differences between groups and why the multivariate model does not include them.

Results:

- In the first paragraph describing the sample I do not understand the use of 95% CI.
We are, indeed, descriptive and it was not relevant to bring this precision. The repetitive use of 95% CI is now omitted.

- Table 1 can be simplified. The laboratory data do not provide too much relevant information (value creating a simplified table and putting the full table as Supplementary Material)
How do you define moderate and severe disease? Why have these categories been chosen to present the data of the sample? It only makes sense to divide the table into these groups if the endpoint would include a comparison between these groups.
In the Methods section, we have now detailed the definition of disease severity according to WHO, as requested previsouly. We also clearly stated our primary and secondary endpoints.  The first Table is now simplified according to the reviewer’s advice.

- The presentation of results is cumbersome and difficult to interpret. I recommend summarizing the presentation of results to generate a clear and direct message.
We have removed anything that weighted the main results. The remaining elements are relevant to the potential reader. We hope that this new version will satisfy the reviewers.

Discussion:

- The AUR found with the model is low. The usefulness of the model should be discussed and discussed against previous literature in greater depth.
As advised by both Reviewers #1 and #2, we further discussed these points, in particular by adding the reference suggested by the first reviewer, a report by Cai et al.

- Limitations should be expanded on, mainly focusing on the bias due to the retrospective data collection (it has been named), the inclusion of only hospitalized patients, and any data on other confounding variables, such as bacterial coinfections?
Indeed, we are thankful for this raised point. Limitations were expanded on accordingly, focusing on the bias due to the retrospective data collection. We added the following details:
“The retrospective nature of the study did not allow us to exhaustively control the existence of other confounding factors such as bacterial co-infections, which could have allowed us to further refine our results.”

Minor concerns

- The introduction should be improved to focus on the problem to be solved and what this work contributes in this field.

We thank the reviewer for this suggestion. We have improved the Introduction to better articulate the problem to be solved. It seems to us relevant to include a section on pathophysiology in a research project on biochemical parameters such as NLR. We hope our Introduction section is now more cohesive and clear. We are certainly ready to aditionnaly remodel this section more thoroughly if the editor wishes us to.

- The English and style should be revised.
A native English speaker has proofread this version of our work.

Reviewer 3 Report

Dear colleague

Vuillaume et al propose the manuscript entitled « Neutrophil to lymphocyte ratio and early variation of NLR to predict in- hospital mortality and severity in ED patients with SARS-CoV-2 infection” for publication in JCM. It is a multicentric retrospective study performed on 1035 patients at the emergency department.

The rationale is nicely described in the introduction and brings the hypothesis that NLR could be interesting for COVID patients.

The primary endpoint is clearly defined as the prognostic value of NLR on in hospital mortality.

Several questions are however important to answer:

Major

  • Severe disease is not clearly defined in the method section, admission to the ICU clearly depends on the local ressources, clear cut criteria need to be used. This is the most important point regarding the fact that the complete analysis is based on this classification.
  • Why were creatinine, lactate, all clinical data at presentation removed from table 1? And creat from the multivariate?

Minor

  • Concerning the excluded patients in ambulatory care, did the authors evaluated the number of re-hospitalization?
  • The flowchart is duplicated on my version which may be an error of the PDF folder
  • 2 interesting studies should be mentioned:
    • Fu et al, Thromb Res 2020 ; 192:3-8
    • Ye et al Respir Res 2020 ; 169 
  • Moreover a larger clinical trial is planned: Ma et al, Crit Care 2020 24:288

Conclusion: this is a very interesting study which potentially add more arguments to support the use of NLR, the main bias in the current form is the definition of severe forms. Why don’t the authors simply divide their population in ventilated (MV, high flow, Non invasive) and not ventilated? This could be an easy way to have objective criteria independent of local ressources.

Author Response

please see tje attachment

Round 2

Reviewer 1 Report

The reviewer thanks the authors for addressing the concerns. Clarification of how patients for discovery and validation data sets were chosen may improve interpretation of the findings. In addition, It would be important to compare the cohort of selected patients with the rest of patients in terms of clinical parameters.

Author Response

Reviewer 1

Yes

Can be improved

Must be improved

Not applicable

Does the introduction provide sufficient background and include all relevant references?

( )

(x)

( )

( )

Is the research design appropriate?

( )

(x)

( )

( )

Are the methods adequately described?

( )

(x)

( )

( )

Are the results clearly presented?

(x)

( )

( )

( )

Are the conclusions supported by the results?

(x)

( )

( )

( )

Comments and Suggestions for Authors

The reviewer thanks the authors for addressing the concerns. Clarification of how patients for discovery and validation data sets were chosen may improve interpretation of the findings. In addition, it would be important to compare the cohort of selected patients with the rest of patients in terms of clinical parameters.

We thank reviewer 1 for their additional suggestions. Both the training and validation cohorts were constituted by random draw; this is now clarified in the Method section. Moreover, in order to better understand the clinical parameters, the tables of these two cohorts were added to the supplemental data.

Reviewer 2 Report

I thank the authors for their efforts in answering all the questions. However, some of the main concerns regarding the methodology employed still remain, diminishing the value of the conclusions from my point of view. The construction of a predictive model entails the use of a strict and specific methodology, which is not specified at any point. The authors indicate in their responses that they have used a step-by-step method that is not specifically defined in the Methods. The split of the sample to perform a validation of the results is not the best strategy, although the sample size is acceptable. Instead, it would be preferable to use validation methods such as bootstrapping. The authors indicate that since the curves of the training and validation cohorts overlap, their results are established, but they do not present the results of these analyses. The authors indicate that they are presented in Figure 3, but this figure appears to present the mortality data as opposed to Figure 2 which presents the severe COVID data. The presentation of univariate analysis results is of no value. The objective of the paper is to present adjusted data, so I consider that the reader is not interested in these unadjusted data and they add complexity to the reading of the paper.
The authors define the ΔNLR as the difference between NLR (NLR H-24 - NLR at admission). I do not understand therefore the figures expressed in the tables (example in global cohort: NLR=5.4, NLR H-24=5, why the ΔNLR=334? 
From my point of view the interpretation of the results remains cumbersome, and the work has too many limitations to consider its conclusions valid.

Reviewer 3 Report

The authors changed and corrected the manuscript which now reach the standards for publication.